# Sex Differences in Anthropometric and Physiological Profiles of Hungarian Rowers of Different Ages

**DOI:** 10.3390/ijerph19138115

**Published:** 2022-07-01

**Authors:** Robert Podstawski, Krzysztof Borysławski, Zsolt Bálint Katona, Zoltan Alföldi, Michał Boraczyński, Jarosław Jaszczur-Nowicki, Piotr Gronek

**Affiliations:** 1Department of Tourism, Recreation and Ecology, University of Warmia and Mazury in Olsztyn, 10-719 Olsztyn, Poland; 2Institute of Health, Angelus Silesius State University, 58-300 Wałbrzych, Poland; kboryslawski@puas.pl; 3Doctoral School of Health Sciences, Faculty of Health Sciences, University of Pécs, 7622 Pécs, Hungary; zsbkatona@googlemail.com (Z.B.K.); zoltan.alfoldi85@gmail.com (Z.A.); 4Department of Health Sciences, Collegium Medicum, University of Warmia and Mazury in Olsztyn, 10-561 Olsztyn, Poland; michal.boraczynski@gmail.com; 5Laboratory of Healthy Aging, Department of Dance, Poznań University of Physical Education, 61-871 Poznań, Poland; gronek@awf.poznan.pl

**Keywords:** rowing, dimorphism index, body composition, motor performance, categories

## Abstract

The aim of this study was to determine sexual differentiation in the anthropometric and physiological characteristics of Hungarian rowers in different age categories. These characteristics were measured for 15–16-year-old juniors (55 men and 36 women), 17–18-year-old older juniors (52 men and 26 women), and 19–22-year-old seniors (23 men and 8 women). The degree of sexual dimorphism was expressed in units of measurement as percentages and the dimorphism index. In all age categories, females had significantly higher body fat indices. Body fat percentage was determined by electrical impedance and by the Pařízková formula, BMI, and skinfold thicknesses. Males had significantly higher body mass, body height, skeletal muscle mass, sitting height, arm span, lower limb length, and body surface area. Males also scored significantly higher values for the following physiological characteristics: peak power, relative peak power, ErVO_2max_, jump height, speed max, force max, and relative maximal power. Analysis of anthropometric and physiological characteristics in Hungarian rowers revealed that sexual dimorphism tended to increase with age, regardless of whether it was expressed in units of measurement, percentages, or dimorphism index values. The age-related increase in the sexual dimorphism of Hungarian rowers suggests that training methods should be carefully selected to accommodate the needs of various age and gender groups.

## 1. Introduction

Sex differences in motor performance have attracted considerable attention over the last 40 years. Sexual dimorphism can be generally defined as morphological and physiological differences between males and females of the same species, and this distinction can be based on differences in size, shape, stature, cranial and facial features, muscularity, strength, and speed [1]. The majority of studies on sex differences have focused on running performance (e.g., [2,3,4,5,6,7,8,9,10,11,12,13]), followed by swimming [14,15], cycling [16], or triathlon [17,18], but few have analysed these differences in the contexts of rowing.

Partly due to anatomical and physiological sex differences, men generally exhibit higher levels of motor performance than women. The muscular strength of women is typically 40–75% of that of men [19]. Men have more muscle mass than women [15,20], which is the main factor underlying gender differences in maximal strength [21]. Men are also more powerful than equally trained women [22], they have a higher maximum oxygen consumption [23], and they demonstrate greater biomechanical efficiency [24]. In terms of power per kg of body mass, sex differences are still evident [25], and the difference in absolute strength between sexes is more noticeable in the upper body than in the lower body [26]. Women have proportionally more fat mass than men [27,28]. This difference, along with the fact that women typically have a smaller heart, a lower haemoglobin concentration, less muscle mass per unit of body weight, and smaller maximal oxygen uptake (VO_2max_), explains the faster performance of men in distance running events [5,6,7,29]. Although several studies have indicated that sex differences in strength may be attributable to lean body mass (LBM), they have also reported that sex differences in power performance were still apparent regardless of body composition and muscle mass [30,31]. With regard to fixed-seat rowers (traditional rowing), Penichet-Tomas et al. [32] demonstrated that in the group of the analysed variables, performance was most highly correlated with body height in male rowers and with muscle mass in female rowers. The cited authors argued that athletic success is more likely to be determined by high lean body mass and a favourable power-to-body mass ratio than by high body mass, whereas high body mass and high BMI have a detrimental effect on performance. Similar observations were made by Winkert et al. [33]. However, there have been no papers comparing the two sexes in terms of anthropometric characteristics in rowing while taking age categories into account.

In addition to differences in muscular strength and power, the physiological reasons for sex differences in motor performance are also attributable to differences in VO_2max_, movement economy, and the exercise intensity at which a high percentage of VO_2max_ can be maintained [7,14]. The ability of men to consume more oxygen per unit of body weight than women appears to be the primary factor underlying sex differences in endurance running motor performance [34,35,36,37] and in rowing [38,39,40]. To complement the existing information on the degree of sexual dimorphism in rowing, it would be interesting to obtain hitherto missing information on the physiological characteristics of male and female rowers, such as relative peak power, jump height, speed max, and relative maximal power.

The difference in power output between women and men ranges from 20–30% for running and speed skating to approximately 45% for swimming, which is consistent with the differences in lower- and upper-body muscle mass and maximal strength between the sexes [20,41,42]. However, to the best of the authors’ knowledge, such information is missing for the sport of rowing, which involves substantial contributions from both upper- and lower-body muscles.

Obtaining additional information on the sexual dimorphism of rowers provides an opportunity to compare this sport with others such as those presented above. Interestingly, sex differences in physiology may affect swimming performance differently than they affect performance in some other sports, and these specific swimming differences may have some relevance to rowing. For example, women are more energy-efficient than men during extreme endurance swimming because they experience less drag [36,37,43,44]. Thus, in non-weight-bearing sports like swimming and rowing, sex differences in performance that are due to physiology may be less evident than those that are observed during weight-bearing exercises (e.g., running) [45]. Competitions (regattas) are usually held over a distance of 2000 m, which corresponds to about 5.5–8 min to finish the racecourse and means that rowing is a strength-endurance sport [46]. This distinguishes rowing from other sports disciplines, and the results regarding sexual dimorphism in this type of effort (hybrid strength-endurance effort) may provide new and interesting information on this topic.

Although the rowing literature contains a good number of papers examining men and women in different age categories (e.g., [47,48]), the number of published articles comparing the two sexes is very small [3,49]. Generally, rowing time has been shown to be slower in female rowers than in male rowers of similar body height and mass [50]. According to Keenan et al. [3], rowing is unique among team sports, and, given the gender shifts in the sport over the past 20 years, it provides an attractive field of research to evaluate predictors that arise from sociocultural conditions and evolved predispositions hypotheses. One reason for the gender shifts in this sport is the rise in popularity of women’s rowing since 1997 when it became a National Collegiate Athletic Association sport, which resulted in an increase in the number of collegiate women’s teams and a corresponding decrease in the number of men’s teams [3]. Therefore, the aim of this study was to determine the sex variation in the anthropometric and physiological characteristics of Hungarian rowers in different age categories. The research hypothesis postulates that age and sex influence the anthropomorphic and physiological parameters of rowers, rowing performance over a distance of 2000 m, and motor test scores.

## 2. Materials and Methods

### 2.1. Participants

The study was conducted in the Gyor rowing club, and the sample consisted of 130 male and 70 female rowers from the seven largest Hungarian rowing clubs. The targeted sampling method was used to select the participants. The following inclusion criteria were applied: rowers from all age groups held valid competition licenses and had participated in national and/or international events over a period of at least one year. In addition, all rowers were required to present a valid medical certificate, they had to train regularly, and their physical activity was not limited (for any reason) to the extent that it would substantially influence their motor fitness levels. All rowers in each club that met these criteria were included in the study.

Each rower was assigned to one of the three age categories: juniors (15–16 years old, 36 women and 55 men), older juniors (17–18 years old, 26 women and 52 men), and seniors (over 18 years old, 8 women and 23 men). The senior groups were relatively young, and the oldest senior rower was only 22. The rowers’ training programs were consistent with the guidelines of the Hungarian Rowing Federation Training Plan: 12–13 h/week for 15- to 16-year-olds, 14–15 h/week for 17- to 18-year-olds, and 16–17 h/week for 19- to 22-year-olds. The aerobic-to-anaerobic training ratio in the above groups was 80:20%, 75:25%, and 70:30%, respectively. The study took place over three consecutive days in the middle of the racing season (8 days after one rowing regatta and 7 days before the next rowing regatta).

The study was consistent with the guidelines and recommendations of the Health Science Council, the Hungarian Scientific and Research Ethics Committee (IV/3067-3/2021/EKU), and the Declaration of Helsinki. The participants received comprehensive information about the research objective, the relevant risks, the applied methods of measurement, and the techniques that would be used in motor tests. These techniques could be practiced directly before the study. The rowers agreed to participate in the study on a volunteer basis by signing informed consent forms.

### 2.2. Procedures, Data Collection and Equipment

#### 2.2.1. Procedures

Each rower was subjected to anthropometric and physiological tests in the middle of the 2020 racing season. On the first day, anthropometric features were measured; on the second, the athletes performed motor tests; and on the third, they covered a distance of 2000 m.

The rowers’ coaches assisted with the measurements. All coaches were instructed not to engage the subjects in any strenuous training on the day before testing. Each subject was always tested in the morning after eating a light meal (800–1200 kcal) containing mainly carbohydrates (60–70%) at least 3–4 h before the study [51]. To measure body height to the nearest 1 mm, a calibrated Soehlne Electronic Height Rod 5003 (Soehnle Professional, Backnang, Germany) was used according to standardized guidelines. To determine body mass (measured to the nearest 0.1 kg), BMI, and body composition characteristics by bioelectrical impedance, including body fat percentage (BFP) and skeletal muscle mass (SMM), an InBody 720 body composition analyser was employed. For the remaining anthropometric characteristics, such as arm span [cm], limb length [cm], sitting height [cm], and BSA [m^2^], the Weiner and Lourie methods [52] were used. To obtain skin fold measurements (abdomen, thigh, lower leg, biceps, triceps, scapula, suprailiac), a Harpenden calliper was used.

#### 2.2.2. Estimation of Relative Body Fat Content

For a calipermetric estimation of relative body fat content, the method developed by Pařízková [53] was used. This procedure is based on measuring 5 skinfold thicknesses: over the biceps and triceps, subscapular, suprailiac, and medial calf. After the sum of the 5 skinfold values is multiplied by 2, the product is then used to find the estimated relative body fat content in a table.

#### 2.2.3. Countermovement Jump Test

To measure the height attained by the center of body mass and the power output of the lower extremities during vertical jumps, a PJS-4P60S force plate (“JBA” Zb. Staniak, Poland) with a 400 Hz sampling rate [31,54,55] was employed. MVJ v.3.4 software (“JBA” Zb. Staniak, Poland) was used to connect the force plate to a PC, and an A/D converter connected the amplifier to a PC. For calculations, the rower’s body mass was treated as a point affected by the force of gravity acting on the body and the vertical component of the platform’s reactive force. Three countermovement jumps (CMJ) were performed by each subject with maximal force. To complete the CMJ test, the subjects performed a vertical jump from a standing erect position, preceded by a countermovement of the upper limbs and lowering of the center of body mass before take-off. The CMJ tests were used to measure maximal force [N] and the rate of displacement [m/s], which provided the basis for determining jump height [cm] (by integrating ground reaction forces) and peak power [W]. Relative peak power [W/kg] was calculated based on body mass.

#### 2.2.4. 2000 m Maximal Rowing Ergometer Test

The participants performed all-out 2000 m tests on a certified rowing ergometer (Concept 2 D). The screen of the ergometer was set to display the number of meters remaining, the average 500 m time, and the accumulated time.

The power output in watts (W) was measured over 2000 m. The calculation of watts was performed as follows: First, the distance was defined: distance = (time/number of strokes) × 500. In the next step, the concept of a “split” was clarified: split = 500 × (time/distance). The watts were calculated as 2.8/(split/500). There were slight differences in intensity due to individual changes in stroke value and the ability to keep the 500 m split time constant. Before the tests, the participants warmed up for 6 min over a 500 m distance, then rested for 6 min, during which time they performed stretching exercises. The estimated relative aerobic capacity (ErVO_2_) was calculated by using the formulas of McArdle et al. [56]: for women the formula is ErVO_2_ = (Y × 1000)/BM, where BM is body mass, and Y = [BM < 61.36 kg; 14.61 − (1.5 × time)]; BM => 61.3 kg; 14.6 − (1.5 × time)]; for men it is ErVO_2_ = (Y × 1000)/BM, where BM is body mass, and Y = [BM < 75 kg; 15.1 − (1.5 × time)]; BM => 75 kg; 15.7 − (1.5 × time)]. The power produced over 2000 m was divided by body weight to obtain the relative performance (rW 2k).

Due to time and logistical constraints, including the need to perform a relatively large number of measurements over three consecutive days, and the desirability of minimizing disturbances to the athletes’ training and changes in their condition, this study did not examine heart rates (HR) and indicators of acid-base balance, such as the lactic acid concentration in the blood, alkaline deficiency or excess, blood pH, and current molecular pressure of CO_2_.

### 2.3. Calculations and Statistical Analysis

For all studied male and female characteristics, basic statistical measures (e.g., mean, standard deviation) were calculated and the normality of the distributions was assessed. Since the distributions did not differ significantly from normality (Shapiro–Wilk test), the Student’s *t*-test was used to assess differences between the men and women. The results were regarded as statistically significant at *p* < 0.05. Additionally, the value of the Szopa dimorphism index was calculated, as given in Podstawski et al. [57]:$$ID=\frac{2\left(x_{m}-x_{f}\right)}{S_{m}+S_{f}}$$
where:
*x_m_*—arithmetic mean of male students in a given age group,*x_f_*—arithmetic mean of female students in a given age group,*S_m_*—standard deviation of male students in a given age group,*S_f_*—standard deviation of female students in a given age group,


Note that differences in measurements (e.g., kg, N, W) between the sexes are presented as male value minus female value, whereas percent differences between the sexes were calculated assuming that the mean value for the male trait was the baseline of 100%.

## 3. Results

The results were presented in tabular form, separately for each age group: 15–16-year-olds (Table 1), 17–18-year-olds (Table 2), and 19–22-year-olds (Table 3). Overall, all analysed age groups showed statistically significant sex differences with respect to the analysed anthropometric and physiological characteristics except for BMI values. additionally, the differences between sexes tended to increase with age, whether expressed in terms of the units of measurement, percentages, or DI values. More specific analyses of each age group are presented below.

### 3.1. Analysis 1: Anthropometric and Physiological Characteristics of 15–16-Year-Old Rowers

The range of sexual dimorphism with respect to the analysed characteristics of the 15–16-year-old male rowers is presented in Table 1. In terms of anthropometric characteristics, the 15–16-year-old female rowers had significantly lower values of the following (the differences in measurements were calculated by subtracting the female value from the male value; male values were the baseline for calculating relative differences): body height (12.1 cm, −6.8%), body mass (5.7 kg, −8.6%), SMM (7.8 kg, −18.5%), sitting height (4.1 cm, −4.5%), arm span (12.9 cm, −7.1%), lower limb length (5.2 cm, −5.1%), and BSA (0.3 m^2^, −5.4%). In terms of physiological characteristics in all motor tests, the 15–16-year-old female rowers achieved significantly slower or lower performances than their male peers: peak power (68.5 W, −27.3%), RPP (0.75 W/kg, −20.0%), time 2000 m (−0.83 min, 11%), ErVO_2max_ (13.9 mL/kg/min, −20.9%; 4.4 L/min, −27.7%), jump height (7.25 cm, −20.1%), speed max (0.3 m/s, −11.5%), force max (269.1N, −17.3%), and RPM (8.01 W/kg, −16.5%) (Table 1). In contrast, the women significantly exceeded the men in terms of body fat as indicated by the following indices: BFP (−11.4%, a relative difference of 92.2% from the males), BFP PF (−7.4%, a relative difference of 32.1%) and measured skinfolds (biceps, triceps, scapula, suprailiac, abdomen, thigh, lower leg), which all displayed values elevated from 19.1 to 52.0% over those of the males (Table 1).

### 3.2. Analysis 2: Anthropometric and Physiological Characteristics of 17–18-Year-Old Rowers

As above, all the analysed characteristics in the 17–18-year-old age group differed significantly between the genders, except for BMI (Table 2). The 17–18-year-old female rowers were significantly shorter (12.8 cm, −7%), lighter (7.8 kg, −10.5%), and recorded lower values for SMM (9.72 kg, −22.5%), sitting height (4.7 cm, −5.0%), arm span (16.1 cm, −8.6%), lower limb length (4.1 cm, −4.0%), and BSA (0.31 m^2^, −16.8%). In contrast, the female rowers had larger values for body fat percentage (12.5%, 97.7% difference relative to the males), skin fold thicknesses (ranging from 23.3 to 71.1% larger), and body fat PF (9.6%, 43.8% difference relative to the males).

In terms of physiological characteristics, the females recorded lower or slower performances in these tests: peak power (113.9 W, −34.8%), RPP (1.2 W/kg, −27.0%), time (−1.0 min, 15%), ErVO_2max_ (15.1 mL/kg/min, −20.5%; 5.4 L/min, −28.7%), jump height (12.7 cm, −31.3%), speed max (0.49 m/s, −17.8%), force max (350.8 N, −20.4%), and RPM (13.5 W/kg, −25.7%) (Table 2).

### 3.3. Analysis 3: Anthropometric and Physiological Characteristics of 19–22-Year-Old Rowers

In this age category, the pattern of statistically significant differences between the sexes was very similar to those observed in the other two categories (Table 3). The women had lower values of the following anthropometric characteristics: body height (13.4 cm, −7.2%), body mass (10.0 kg, −12.3%), SMM (9.5 kg, −23.2%), sitting height (5.5 cm, −5.7%), arm span (13.1 cm, −6.9%), lower limb length (7.0 cm, −6.8%), and BSA (0.4 m^2^, −8.6%). In turn, the women recorded larger values for body fat percentage (−13.5%, relative difference of 80.8%), skin fold thicknesses (ranging from 44.3 to 79.8%), and body fat PF (−10.6%, relative difference of 47.1%). In terms of physiological characteristics, the females recorded lower or slower values: peak power (117.5, −31.6%), RPP (1.0 W/kg, −22.2%), time (−0.89 min, 13.5%), ErVO_2_ max (9.1 mL/kg/min, −12.5%; 5.8 L/min, −22.3%), jump height (10.1 cm, −26.1%), speed max (04 m/s, −13.9%), force max (325.4 N, −17.9%), and RPM (10.5 kg, −21.2%) (Table 3).

## 4. Discussion

The purpose of this study was to assess the sex differences between male and female Hungarian rowers of different ages. We hypothesized that the anthropometric and physiological characteristics of the female and male rowers would differ significantly and that these characteristics would also differ between age categories. The results obtained in this study are consistent with the research hypothesis.

### 4.1. Sex Differences in Anthropometric Characteristics

This study showed that there were significant differences in anthropometric characteristics between the sexes in all age categories, as reflected by the values of the dimorphism index (DI). Male Hungarian rowers had greater heights and weights than their female counterparts, and the differences between the sexes increased with age (e.g., for body height, DI values increased from 1.6 to 2.9; for body mass, from 0.6 to 1.3). These differences are likely to influence rowers’ performance. Studies evaluating the anthropometric characteristic of female and male adult rowers (body mass and height) have emphasized the significance of body mass and height [58,59,60,61,62] and have demonstrated that body size and proportions [63,64,65,66] are important predictors of success in international-level rowing. Similar relationships have also been found between these characteristics and the rowing performance of juniors [59,67,68].

Generally speaking, BFP and BFP PF differed the most between the sexes in each age category as shown by the respective DI values (in terms of increasing ages: −2.0, −2.1, −2.7; and −1.8, −2.9, −3.3), and these differences were highly statistically significant (*p* < 0.001). Interestingly, the difference in body fat was largest among the seniors (19–22 years), and the range of BFP in the male rowers was much wider than the 6 to 10% BFP observed in elite males by Hagerman et al. [69]. Differences in BFP that depend on sex are typical in athletes [70]. Although excess body fat can impair rowing performance, the importance of BFP in rowing compared to other sports is not entirely clear. Generally speaking, increased body mass, characterised by a high BFP and BMI, adversely affects 2000 m rowing ergometer performance [33,71,72,73], and increased muscle mass with a high lean body mass and a favourable power-to-body mass ratio are predictors of success in rowing [32,66]. Similarly, in studies designed to determine the best performance predictive parameters [72,73], fat-free mass emerged as one of the strongest correlates with performance. These findings could explain the trend toward lower BFP in elite rowers observed by Mikulić [67], and most experts concur that proper proportions of tissue components are important for rowers, along with low body fat content and high fat-free mass [60]. According to Yoshiga and Higuchi [74], this may be the case because an association between the fat-free mass and blood volume and stroke volume of the heart has been established (i.e., greater fat-free mass is associated with higher aerobic capacity, which is crucial for successful rowing performance). However, Majundar et al. [65] found a positive correlation between rowing performance and body fat percentage and noted that a certain amount of fat is required for the maintenance of body metabolism, although excess adiposity negatively influences performance [65]. In rowing, moreover, the body mass is typically supported by a sliding seat, and body fat in rowers does not put them at the same disadvantage as athletes who carry their own body weight (e.g., runners, jumpers, etc.).

In contrast to BFP and BFP PF, BMI differed the least between the sexes in terms of the DI, and these differences decreased as the age of the rowers increased (15–16 years: −0.49, 17–18 years: −0.36, 19–22 years: −0.27). The mean BMI values in the oldest age categories (men—23.6 kg/m^2^, women—24 kg/m^2^), and particularly that of the women, were close to the upper limit of the normal range and near the value that is considered optimal for rowers, 24 kg/m^2^ [74,75,76]. Forjasz [60] found that World Cup and Olympic Games finalists had not only higher BMI values than non-finalists, but also higher SMM values. Therefore, when reviewing the results regarding the content of BFP and SMM, it is important to mention that, according to some authors, these are among the most important anthropometric determinants that substantially affect rowing performance. This hypothesis is supported in part by research conducted by Pinechet-Tomás et al. [32] among traditional rowers, which indicated that body height is the best performance predictor for male rowers, whereas muscle mass is the best predictor for female rowers, which may be due to women having higher BFP values and lower SMM values. This suggests that body composition, including a high lean body mass, and an adequate power-to-body mass ratios, are better predictors of rowing performance than high body mass. Similar observations were made by Winkert et al. [33] who demonstrated that high body mass and high BMI were negatively correlated with performance. Garrido-Chamorro et al. [77] and Mazić et al. [78] concluded that body composition and fat and muscle tissue distribution in the lower and upper half of the body cannot be reliably estimated based on BMI alone. Thus, BMI values may not be the best way to assess the sexual dimorphism of rowers. Indeed, the males and females in this study differed very little in this regard although there were important differences between the sexes with regard to SMM and LBM, and among the older juniors, the two largest DI values were for SMM (2.7) and time for covering 2000 m on a rowing ergometer (−2.7).

With regard to sitting height, lower limb length, and arm span, the males recorded higher values, and the DI increased with age. This increase in sexual dimorphism in older rowers has important implications for their rowing performance. For example, Penichet-Tomás et al. [66] found that higher-performing traditional rowers have significantly longer trunk lengths than lower-performing ones, which led them to hypothesize that this is because trunk movement plays a significant role in traditional rowing, which was also the conclusion of Izguierdo-Gabaren et al. [79]. Ng et al. [80] also observed differences in trunk and pelvic kinematics between male and female rowers. Similarly, Li et al. [49] found that female rowers exhibit a greater range of motion in the lumbar spine, thorax, and shoulders than males due to more extended positions at the finish. Additionally, various authors have pointed out that female rowers may use their higher spinal flexibility [80] and possibly their spinal alignment [81] to alter their rowing technique and compensate for their lower body size, muscle strength, level of endurance abilities [49] and maximal oxygen uptake [74]. Longer limbs are also an advantage because they allow more force to be generated during rowing and a longer stroke, as the catch and drive components of the stroke involve all four extremities [82,83]. Longer stroke lengths are closely associated with high-level rowing performance [73].

### 4.2. Sex and Age Differences in Physiological Characteristics

Regarding gender differences in physiological characteristics across all age categories, males outperformed females in all of the motor tests that were used. The highest DI values (given in order of increasing age category) were for peak power (1.8, 2.78, 2.6), then RPP (1.6, 2.8, 2.5) and time needed to complete 2000 m (−1.7, −1.69, −2.43), which was shortest in the senior males (6.6 min). The smallest differences in DI values were for force max (1.04, 1.63, 1.56). Additionally, small DI values were recorded for RPM (1.38) in the juniors, and relative ErVO_2max_ (1.5) and force max (1.6) in the seniors. A study of elite Polish oarsmen (men and women) by Klusiewicz and Faff [48] showed that the HR values obtained by women while covering a distance of 2000 m on a rowing ergometer were higher than those obtained by men (195 ± 6 and 193 ± 8 bpm, respectively), although the maximum values were higher in men (204 and 205 bpm, respectively). Keenan et al. [3] studied rowers and observed that race times across years, weight classes, and finishing places were shorter among male than female athletes. Despite the above, the times relative to the first place at higher finish places were faster among women than men. In both Collegiate and Junior World Rowing Championships, female rowers improved their performance to a greater extent than male athletes between 1997 and 2016. Those authors also found that in the heavyweight class, the drop-off in rowing performance was greater for men than for women, but in the lightweight class, it was smaller for men than for women.

In the present study, the absolute and relative VO_2max_ of the female and male rowers differed significantly between age categories. Interestingly, Klusiewicz et al. [47] found that, in elite Polish crews, Olympic medalists, and World Champions, the VO_2max_ increased markedly as female rowers increased from 20 to 22 years of age and as males increased from 19 to 19.9 and from 21 to 22 years of age, reaching respective values exceeding 4.0 and 6.0 l/min for the females and males, respectively. When considering the percent increase in VO_2max_, it increased by 22.0 and 11.7% in Polish females and males, respectively.

### 4.3. Strengths and Limitations

This study makes a useful contribution to the literature by shedding new light on sexual dimorphism in rowers in three different age categories and across all weight categories, which has heretofore received scant attention. The number of papers dealing with sexual dimorphism in rowing is very low. The strengths of this study include the inclusion of a relatively representative and numerous sample of rowers in three age categories, with the exception of the number of female senior rowers (eight rowers). However, it should be emphasized that we examined all (100%) licensed rowers from seven of the best and largest Hungarian rowing clubs, and these rowers trained in very similar environmental conditions and fulfilled the same selection criteria, which is very important for comparative analyses. It can also be noted that some studies of rowers had similar sample sizes [e.g., 37–N = 10; 84–N = 8; 85–N = 8). The Hungarian rowers in this study are not particularly outstanding; instead, they are typical rowers like those who constitute the vast majority of the rowing community, which makes it possible to make relevant comparisons with regard to sexual dimorphism. This is a novelty because the vast majority of studies have focused on finalists at the Olympics, world championships or other major regattas —such individuals are very few and are spread across continents or countries, making comparisons very difficult. In this study, the differences in sexual dimorphism are presented in the form of numerical values, percentages, and values of the sexual dimorphism index, which is very rare in this type of study, especially when such a large number of anthropometric and physiological measurements were taken. However, the limited amount of time available to conduct this large number of measurements meant that, inevitably, some measurements had to be omitted (HR and acid-base balance indices). Nevertheless, it should be emphasized that, given the limited time and organizational possibilities, a mass study (with relatively large study groups) that includes such additional analyses is very difficult to conduct. Although a certain limitation of the present study may also be the measurement of estimated values of VO_2max_, the scientific literature shows that these measurements are quite commonly used in this type of research [84,85], including research on rowers [48,86].

### 4.4. Future Research

An interesting topic for future research on sexual dimorphism in rowers would be the ratio between the length of the index finger (2D) and the ring finger (4D). This ratio is already formed during the early stages of human foetal development and does not change throughout life. Manning et al. [87] found that sexual dimorphism of this index becomes apparent as early as 2 years of age. The differences are due to exposure to sex hormones present in the amniotic fluid during the prenatal period. Manning et al. [88] considered the 2D:4D ratio a prenatal biomarker that determines the balance between testosterone and oestrogen levels. Testosterone has a masculinizing effect and stimulates the growth of the fourth finger, while estrogen elongates the second finger, which means that, in men, the index finger is most often shorter than the fourth finger, with a ratio close to 0.98, while in women, this ratio is higher at about 1 [87]. This sexual dimorphism has been noted across Africa, Europe, Jamaica, and Asia in 13 different populations [89]. Low values of the index tend to be possessed by successful male athletes, especially in sports requiring efficient cardiovascular function (soccer, cross-country skiing, middle and long-distance running) [90], although this phenomenon has not been found in women [91]. However, the 2D:4D has not been examined and compared in male and female rowers.

## 5. Conclusions

The results of our study indicate that, from the age of 15 to 22 years, there are significant sex differences in anthropometric and physiological characteristics in Hungarian rowers. The women have significantly larger values for body fat (BFP, BFPPF, BMI, and skinfold thicknesses), while the men have significantly larger values for body mass, body height, SMM, sitting height, arm span, lower limb length, and BSA. Regarding physiological characteristics, male rowers significantly outperform female rowers in all motor tests, including peak power, RPP, ErVO_2max_, jump height, speed max, force max, and RPM. Moreover, the DI values for the analysed anthropometric and physiological characteristics increase with age. The age-related increase in the sexual dimorphism of Hungarian rowers suggests that training methods should be carefully selected to accommodate the needs of various age and gender groups.

## Figures and Tables

**Table 1 ijerph-19-08115-t001:** Sexual dimorphism of anthropometric, physiological, and motor characteristics in rowers aged 15–16 years and statistical significance of differences.

Characteristic		Age Category 15–16 [Years]										
		Men			Women			Differences				
		Mean	SD	Min-Max	Mean	SD	Min-Max	M–F	%	DI	*t*	*p*
Body height [cm]		178.70	7.22	162.1–193.4	166.63	7.64	156.7–187.1	12.08	−6.8	1.63	7.63	<0.001
Body mass [kg]		66.39	10.89	39.6–91.5	60.70	7.08	49.2–76.4	5.69	−8.6	0.63	2.78	0.007
Body fat [%]		12.39	5.54	4.0–28.9	23.81	5.73	13.9–32.1	−11.42	92.2	−2.03	9.17	<0.001
SMM [%]		41.90	5.04	14.1–52.6	34.15	2.88	29.0–41.3	7.75	−18.5	1.96	8.24	<0.001
BMI [kg/m^2^]		20.71	2.68	15.02–28.31	21.86	2.01	18.81–26.42	−1.15	5.5	−0.49	2.19	ns
Sitting height [cm]		92.50	4.60	79.4–100.3	88.38	3.89	83.1–100.0	4.12	−4.5	0.97	4.43	<0.001
Arm span [cm]		181.13	13.00	104.3–196.0	168.22	8.08	155.4–188.0	12.91	−7.1	1.22	5.32	<0.001
Lower liIght [cm]		101.01	4.07	92.1–111.0	95.86	6.10	85.4–112.4	5.15	−5.1	1.01	4.81	<0.001
BSA [m^2^]		1.67	0.36	0.89–3.08	1.41	0.22	1.07–1.94	0.26	−5.4	0.90	3.89	<0.001
Skin fold thickness [mm]	Biceps	7.04	3.57	2–20	10.69	4.07	3–22	−3.66	52.0	−0.96	4.52	<0.001
	Triceps	14.16	5.66	5–29	18.89	4.73	10–29	−4.73	33.4	−0.91	4.15	<0.001
	Scapula	10.96	4.57	4–31	14.69	4.31	8–24	−3.73	34.0	−0.84	3.89	<0.001
	Suprailiac	9.95	5.66	4–33	14.33	4.50	6–24	−4.39	44.1	−0.86	3.91	<0.001
	Abdomen	14.06	6.89	5–42	17.31	6.65	8–36	−3.25	23.1	−0.48	2.23	0.028
	Thigh	20.36	7.84	6–46	24.25	7.19	10–38	−3.89	19.1	−0.52	2.39	0.019
	Lower leg	14.07	6.27	4–30	16.86	5.07	6–25	−2.78	19.8	−0.49	2.20	0.030
BFP ^PF^ [%]		23.03	4.04	13.8–31.5	30.41	4.10	22.9–36.5	−7.38	32.1	−1.81	8.22	<0.001
Peak power 2000 m [W]		250.55	44.60	138–322	182.09	30.12	129–246	68.46	−27.3	1.83	7.83	<0.001
RPP 2000 m [W/kg]		3.76	0.53	2.11–4.71	3.01	0.42	2.25–3.73	0.75	−20.0	1.58	6.92	<0.001
Time 2000 m [min] *		7.51	0.51	6.85–9.09	8.34	0.47	7.50–9.30	−0.83	11.0	−1.69	7.57	<0.001
ErVO_2max_ [mL/kg/min]		66.43	9.49	38.32–82.47	52.52	9.98	30.59–67.70	13.91	−20.9	1.43	6.48	<0.001
ErVO_2max_ [L/min]		4.41	0.77	2.06–5.42	3.19	0.71	1.75–4.45	1.22	−27.7	1.65	7.39	<0.001
Jump height [cm]		36.02	4.97	23.5–44.4	28.77	4.61	20.7–37.6	7.25	−20.1	1.51	7.00	<0.001
Speed max [m/s]		2.59	0.19	2.06–2.91	2.29	0.21	1.89–2.65	0.30	−11.5	1.49	7.01	<0.001
Force max [N]		1551.35	323.58	899–2317	1282.25	194.70	950–1916	269.10	−17.3	1.04	4.48	<0.001
RPM [W/kg]		48.43	5.69	34.8–60.9	40.42	5.94	30.1–52.7	8.01	−16.5	1.38	6.45	<0.001

Notes: ns—non-significant difference (*p* > 0.05), ^PF^—Pařízková’s formula, SMM—skeletal muscle mass, RPP—relative peak power, RPM—relative maximal power. M–F—difference (men minus women), %—percent difference between men and women (male value is baseline, i.e., 100%), DI—Szopa dimorphism index, *—shorter time is a better result.

**Table 2 ijerph-19-08115-t002:** Sexual dimorphism of anthropometric, physiological, and motor characteristics in rowers aged 17–18 years and statistical significance of differences.

Characteristic		Age Category 17–18 [Years]										
		Men			Women			Differences				
		Mean	SD	Min-Max	Mean	SD	Min-Max	M–F	%	DI	*t*	*p*
Body height [cm]		183.02	7.27	167.7–197.4	170.21	6.74	160.0–187.4	12.81	−7.0	1.83	7.51	<0.001
Body mass [kg]		73.70	8.43	56.6–89.7	65.95	7.85	53.2–84.1	7.75	−10.5	0.95	3.91	<0.001
Body fat [%]		12.84	5.39	5.3–33.0	25.37	6.68	8.3–35.3	−12.54	97.7	−2.08	8.89	<0.001
SMM [%]		43.30	3.47	27.2–49.2	33.57	4.48	28.0–47.4	9.72	−22.5	2.45	10.52	<0.001
BMI [kg/m^2^]		21.98	2.10	18.28–29.47	22.74	2.15	18.48–27.19	−0.76	3.4	−0.36	1.49	ns
Sitting height [cm]		95.33	3.56	87.5–105.1	90.60	3.61	85.4–99.9	4.74	−5.0	1.32	5.51	<0.001
Arm span [cm]		188.43	8.56	168.5–203.0	172.30	7.72	159.5–192.0	16.13	−8.6	1.98	8.10	<0.001
Lower limb length [cm]		102.40	4.98	90.3–111.4	98.35	5.89	87.9–112.5	4.06	−4.0	0.75	3.19	0.002
BSA [m^2^]		1.88	0.26	1.32–2.31	1.56	0.23	1.21–2.19	0.31	−16.8	1.27	5.18	<0.001
Skin fold thickness [mm]	Biceps	5.69	3.13	3–21	9.73	3.34	5–17	−4.04	71.1	−1.25	5.24	<0.001
	Triceps	12.08	4.50	5–26	18.85	4.97	10–31	−6.77	56.0	−1.43	6.02	<0.001
	Scapula	9.96	3.16	6–23	15.27	4.41	9–23	−5.31	53.3	−1.40	6.08	<0.001
	Suprailiac	8.26	3.76	4–21	13.15	4.40	5–25	−4.90	59.3	−1.20	5.11	<0.001
	Abdomen	12.29	4.81	5–26	15.15	3.93	7–22	−2.86	23.3	−0.65	2.62	0.011
	Thigh	18.45	7.54	7–39	26.42	5.69	14–38	−7.97	43.2	−1.21	4.74	<0.001
	Lower leg	12.31	5.99	5–30	16.77	4.28	10–25	−4.46	36.2	−0.87	3.37	0.001
Body fat ^PF^ [%]		21.86	4.14	14.5–30.9	31.44	2.52	26.5–36.9	−9.58	43.8	−2.88	10.63	<0.001
Peak power 2000 m [W]		326.80	54.48	210–435	212.92	27.85	155–261	113.88	−34.8	2.77	9.62	<0.001
RPP 2000 m [W/kg]		4.42	0.51	3.10–5.31	3.23	0.36	2.35–4.01	1.20	−27.0	2.76	10.27	<0.001
Time 2000 m [min] *		6.87	0.41	6.20–7.90	7.90	0.36	7.35–8.75	−1.03	15.0	−2.70	10.55	<0.001
ErVO_2max_ [mL/kg/min]		73.44	6.31	56.69–88.19	58.37	6.82	41.08–73.26	15.07	−20.5	2.30	9.34	<0.001
ErVO_2max_ [L/min]		5.40	0.61	3.84–6.40	3.85	0.53	2.58–4.67	1.55	−28.7	2.72	3.91	<0.001
Jump height [cm]		40.59	7.62	24.7–58.9	27.90	3.10	21.6–33.7	12.69	−31.3	2.37	8.13	<0.001
Speed max [m/s]		2.74	0.29	1.97–3.33	2.25	0.13	1.97–2.49	0.49	−17.8	2.31	8.16	<0.001
Force max [N]		1721.19	283.77	1180–2712	1370.39	145.40	1124–1690	350.81	−20.4	1.63	5.91	<0.001
RPM [W/kg]		52.41	7.88	30.2–63.4	38.95	3.86	30.6–45.7	13.46	−25.7	2.29	8.21	<0.001

Notes: ns—non-significant difference (*p* > 0.05), ^PF^—Pařízková’s formula, SMM—skeletal muscle mass, RPP—relative peak power, RPM—relative maximal power. M–F—difference (men minus women), %—percent difference between men and women (male value is baseline, i.e., 100%), DI—Szopa dimorphism index, *—shorter time is a better result.

**Table 3 ijerph-19-08115-t003:** Sexual dimorphism of anthropometric, physiological, and motor characteristics in rowers aged 19–22 years and statistical significance of differences.

Characteristic		Age Category 19–22 [Years]										
		Men			Women			Differences				
		Mean	SD	Min-Max	Mean	SD	Min-Max	M–F	%	DI	*t*	*p*
Body height [cm]		184.96	4.98	174.4–194.0	171.58	4.14	166.2–179–8	13.39	−7.2%	2.94	6.81	<0.001
Body mass [kg]		80.75	8.09	62.1–91.2	71.19	6.49	63.2–81.5	9.56	−1.08	1.31	3.01	0.005
Body fat [%]		16.67	4.33	9.4–22.9	30.15	5.52	19.9–36.2	−13.48	80.8	−2.74	7.01	<0.001
SMM [%]		41.12	3.83	29.9–46.6	31.60	4.02	28.0–40.9	9.52	−23.2	2.43	5.95	<0.001
BMI [kg/m^2^]		23.61	2.29	18.28–26.68	24.17	1.84	21.12–26.47	−0.56	2.04	−0.27	0.62	ns
Sitting height [cm]		96.56	2.04	93.9–100.1	91.01	2.01	88.0–93.2	5.54	−5.7	2.74	6.64	<0.001
Arm span [cm]		189.29	5.53	179.7–197.5	176.19	5.56	168.4–185.0	13.10	−6.9	2.36	5.77	<0.001
Lower limb lenght [cm]		103.02	3.98	96.5–113.8	96.01	2.85	93.9–102.4	7.00	−6.8	2.05	4.56	<0.001
BSA [m^2^]		2.09	0.25	1.50–2.63	1.70	0.18	1.52–2.04	0.39	−8.6	1.80	4.05	<0.001
Skin fold thickness [mm]	Biceps	5.96	2.69	3–12	9.88	2.85	6–15	−3.92	65.8	−1.42	3.50	0.002
	Triceps	12.65	4.57	5–20	22.75	3.45	20–29	−10.10	79.8	−2.52	5.69	<0.001
	Scapula	10.70	3.48	3–18	17.75	3.92	12–24	−7.05	66.0	−1.91	4.78	<0.001
	Suprailiac	8.52	2.29	5–13	15.00	1.31	13–17	−6.48	76.0	−3.60	7.52	<0.001
	Abdomen	12.48	3.82	7–22	18.00	4.75	15–29	−5.52	44.3	−1.29	3.31	0.003
	Thigh	18.48	6.01	4–29	31.38	5.26	26–41	−12.90	69.8	−2.29	5.38	<0.001
	Lower leg	13.39	4.46	4–22	20.13	3.23	15–24	−6.73	50.3	−1.75	3.91	<0.001
Body fat ^PF^ [%]		22.43	3.62	12.2–26.6	33.00	2.73	27.9–35.6	−10.57	47.1	−3.33	6.64	<0.001
Peak power 2000 m [W]		372.22	52.96	292–461	254.75	38.24	180–294	117.47	−31.6	2.58	5.75	<0.001
RPP 2000 m [W/kg]		4.59	0.45	3.57–5.35	3.57	0.37	2.79–4.12	1.02	−22.2	2.51	5.82	<0.001
Time 2000 m [min] *		6.56	0.32	6.08–7.08	7.45	0.41	7.07–8.32	−0.89	13.5	−2.43	6.30	<0.001
ErVO_2max_ [mL/kg/min]		72.61	5.59	60.76–84.99	63.53	6.25	49.82–71.91	9.08	−12.5	1.53	3.84	<0.001
ErVO_2max_ [L/min]		5.83	0.46	5.08–6.58	4.53	0.62	3.22–5.10	1.30	−22.3	2.41	6.29	<0.001
Jump height [cm]		38.44	6.31	22.9–51.0	28.39	2.34	25.0–32.9	10.05	−26.1	2.33	4.36	<0.001
Speed max [m/s]		2.66	0.24	2.08–3.09	2.29	0.11	2.09–2.46	0.37	−13.9	2.13	4.21	<0.001
Force max [N]		1814.74	272.22	1328–2548	1489.38	146.00	1319–1708	325.36	−17.9	1.56	3.20	0.003
RPM [W/kg]		49.39	6.04	36.3–63.2	38.91	3.36	32.7–42.4	10.47	−21.2	2.23	4.63	<0.001

Notes: ns—non-significant difference (*p* > 0.05), ^PF^—Pařízková’s formula, SMM—skeletal muscle mass, RPP—relative peak power, RPM—relative maximal power. M–F—difference (men minus women), %—percent difference between men and women (male value is baseline, i.e., 100%), DI—Szopa dimorphism index. *—shorter time is a better result.

## Data Availability

The data used to support the findings of this study are restricted by the Ethics Committee of the University of Warmia and Mazury in Olsztyn (UWM), Poland in order to protect the participants’ privacy. The data are available from R.P., E-mail: podstawskirobert@gmail.com for researchers who meet the criteria for access to confidential data.

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
