# Peer review of "Sex Differences in Anthropometric and Physiological Profiles of Hungarian Rowers of Different Ages"

_ijerph, 2022, doi:10.3390/ijerph19138115_

Round 1

Author Response

Dear Reviewer,

I am enclosing a revised manuscript and a response to the reviews.

Sincerely,

Robert Podstawski

Reviewer 2 Report

Many thanks to the authors for their work and for the opportunity to read and evaluate it.

line 23 (remove the statistical analysis)

line 39 (in the opinion of LEPERs???? - withdraw)

The introduction is too extensive and goes beyond the subject of the study, there are many comparisons with other modalities. The authors should redo it and focus more on the topic of study.

Do you consider that 130 man and 70 female is representative of the entire rowing population in Hungary?

line 154 - put %

line 181 - caliper

In the methodology, I propose to put 2.2 and then subsections of 2.2.1/2.2.2/2.2.3/...

Because general section 2.2 lacks procedures, data collection, and material. Must do a general 2.2 (for example) and other specifics.

2.6. P CHARACTERIZATION IS MISSING.

RESULTS

The results are all described before the tables and the reader is lost, as the explanation is quite extensive. should make a brief explanation before or after the table respectively.

All results follow the same trend.

All results follow the same trend, the differences are all significant, only the BMI is not significant and women in each age range have more fat mass.

The discussion, as well as a conclusion, show that this study is not groundbreaking.

The description you found in rowing athletes will be like it in all athletes from other modalities or even in non-athletes, as long as you compare the stages of age and sex.

I do not agree with the strengths of the presented study. I consider that it has more limitations than strengths.

Author Response

Dear Reviewer,

I am enclosing a revised manuscript and a response to the review.

Sincerely,

Robert Podstawski

Reviewer 3 Report

I want to thank the editor for the opportunity to review the manuscript "Sex differences in anthropometric and physiological profiles of Hungarian rowers of different ages". Addressing sex factor in different sport is important and provide impactful information for trainers and all those involved in health science. Despite the idea is not original, the quality of the study and the sample included provide an optimal background to accept this manuscript.
I want to congratulate with the authors for the well-written article and for the well-conducted study, I have no further comments.

Additional Comments:

In the manuscript presented by the authors, a variety of measurements have been provided to describe anthropometric and physiological characteristics of rowers. The sample size looks appropriate and providing different age categories help to tailor future interventions and training considering the effect of age. The only minor consideration, although it is merely a point of view instead of a recommendation, is that when considering adolescents, chronological age is always a possible source of bias and biological age should be preferred. However, the reviewer recognizes that in the practical setting it might be complicated to perform valid measurements of biological age (e.g., hand x-ray or Tanner scale evaluations), therefore the age categories provided are sufficient to describe the groups.

Author Response

(The authors gave the same response as above.)

Round 2

Reviewer 2 Report

the work has been substantially altered and although I consider it to be nothing new, I will propose it for publication